# DNA Methylation in Periodontal Disease: A Focus on Folate, Folic Acid, Mitochondria, and Dietary Intervention

**DOI:** 10.3390/ijms26073225

**Published:** 2025-03-30

**Authors:** Elzbieta Pawlowska, Joanna Szczepanska, Marcin Derwich, Piotr Sobczuk, Nejat Düzgüneş, Janusz Blasiak

**Affiliations:** 1Department of Pediatric Dentistry, Medical University of Lodz, Pomorska 251, 92-216 Lodz, Poland; elzbieta.pawlowska@umed.lodz.pl (E.P.); joanna.szczepanska@umed.lodz.pl (J.S.); marcin.derwich@umed.lodz.pl (M.D.); 2Emergency Medicine and Disaster Medicine Department, Medical University of Lodz, Pomorska 251, 92-209 Lodz, Poland; piotr.sobczuk@umed.lodz.pl; 3Department of Orthopaedics and Traumatology, Polish Mothers’ Memorial Hospital—Research Institute, Rzgowska 281, 93-338 Lodz, Poland; 4Department of Biomedical Sciences, University of the Pacific—San Francisco Campus, San Francisco, CA 94103, USA; nduzgunes@pacific.edu; 5Faculty of Medicine, Collegium Medicum, The Mazovian University in Plock, 04-920 Plock, Poland

**Keywords:** periodontal disease, periodontitis, DNA methylation, mitochondrial quality control, folate, dietary intervention

## Abstract

Although periodontal disease (PD) is reported to be associated with changes in various genes and proteins in both invading bacteria and the host, its molecular mechanism of pathogenesis remains unclear. Changes in immune and inflammatory genes play a significant role in PD pathogenesis. Some reports relate alterations in cellular epigenetic patterns to PD characteristics, while several high-throughput analyses indicate thousands of differentially methylated genes in both PD patients and controls. Furthermore, changes in DNA methylation patterns in inflammation-related genes have been linked to the efficacy of periodontal therapy, as demonstrated by findings related to the cytochrome C oxidase II gene. Distinct DNA methylation patterns in mesenchymal stem cells from PD patients and controls persisted despite the reversal of phenotypic PD. Methyl groups for DNA methylation are supplied by S-adenosylmethionine, which is synthesized with the involvement of folate, an essential nutrient known to play a role in maintaining mitochondrial homeostasis, reported to be compromised in PD. Folate may benefit PD through its antioxidant action against reactive oxygen and nitrogen species that are overproduced by dysfunctional mitochondria. As such, DNA methylation, dietary folate, and mitochondrial quality control may interact in PD pathogenesis. In this narrative/hypothesis review, we demonstrate how PD is associated with changes in mitochondrial homeostasis, which may, in turn, be improved by folate, potentially altering the epigenetic patterns of immune and inflammatory genes in both the nucleus and mitochondria. Therefore, a folate-based dietary intervention is recommended for PD prevention and as an adjunct therapy. At the same time, further research is needed on the involvement of epigenetic mechanisms in the beneficial effects of folate on PD studies.

## 1. Introduction

Chronic periodontal disease (PD), characterized by the presence of at least 10% of sites with clinical attachment loss (CAL) of 4 mm or higher and/or a probing pocket depth (PPD) of at least 3 mm, is driven by the interplay between invading pathogenic microorganisms and the host’s response. After colonization of the periodontal tissue, the host forms functional complexes that respond to pathogen-associated molecular patterns, leading to a cascade of molecular events, including the activation of the complement system, production of cytokines, activation of matrix metalloproteinases, and neuropeptides [1]. Periodontal pathogens also induce the overactivation of the host immune system in the periodontal tissue, resulting in excessive infiltration of immune cells and stimulation of osteoclastic activity. Thus, the host response requires the activity of numerous proteins produced by the expression of corresponding genes, which depends on the host’s genetic constitution and epigenetic factors. The host genome sequence determines this genetic constitution, which is difficult to modify by either the host or invading microorganisms. Changes in many genes have been reported to be associated with periodontal disease (PD) [2]. Any specific DNA sequence associated with PD is considered a risk factor and requires sophisticated treatments, such as genome editing, for reversal. Consequently, pathogenic genetic variants are regarded as non-modifiable risk factors for PD. Conversely, the DNA sequence of a gene serves as a template for gene expression, which is further influenced by epigenetic modifications imposed on that gene. These modifications include DNA methylation, histone modification, and changes resulting from the action of non-coding RNAs (ncRNAs). The pattern of epigenetic modifications, unlike the DNA sequence, can be altered by a variety of external and internal factors and compounds, including those found in the diet. Therefore, if the epigenetic pattern of a gene is associated with the occurrence or progression of PD, it is considered a modifiable risk factor for this disease.

Though the role of microRNAs, recognized by the 2024 Nobel Prize in Physiology or Medicine, in regulating gene expression is becoming clearer, DNA methylation remains the simplest epigenetic modification to monitor and manipulate. The importance of epigenetics in PD has been explored in several studies, yet most concentrate on DNA methylation. Reports indicate that nutritional compounds influence PD pathogenesis, leading to dietary considerations for PD prevention and treatment (as reviewed in [3]). Conversely, the genome’s epigenetic landscape, known as the epigenome, is sensitive to diet. Thus, dietary changes can potentially affect the epigenetic patterns of numerous genes, including those crucial for responding to the threat of periodontal pathogens.

Periodontal disease results from impaired maintenance and repair of the periodontium, which necessitates various enzymatic activities and substrates. Folate deficiency is characterized by interference with the maturation of epithelial cells, impaired keratinization, and increased susceptibility to ulceration, which may affect the gingival epithelium [4]. Folate deficiency is among the most common, if not the most prevalent, nutrient deficiencies worldwide [5].

In this narrative perspective review, we provide fundamental information on DNA methylation and its role in regulating gene expression. We examine experimental results related to DNA methylation and the impairment of mitochondrial quality control in PD. Folate is highlighted as a crucial component of the DNA methylation process and a vital nutrient that is often found to be deficient in PD, playing an important role in mitochondrial quality control. We explore the impact of diet on the prevention and treatment of PD and ultimately conclude with the possibility that dietary interventions may be beneficial for PD patients, as indicated by changes in DNA methylation patterns.

## 2. DNA Methylation: A Key Component of the Epigenetic Regulation of Gene Expression

DNA methylation in humans can occur in various ways, but the methylation that has signaling significance in the epigenetic regulation of gene expression happens through the transfer of a methyl group from S-adenosyl-methionine (SAM) to the 5-carbon of cytosine in DNA, resulting in 5-methylcytosine (5 mC).

DNA methylation is catalyzed by DNA methyltransferases (DNMTs), and humans have at least three such enzymes: DNMT1, DNMT3A, and DNMT3B. DNMT3A and DNMT3B are responsible for the methylation of cytosines in both DNA strands (de novo methylation) (Figure 1). Because DNA polymerase does not distinguish between methylated and non-methylated templates, it inserts C and not 5-mC opposite to thymine in the template strand derived from previously methylated DNA. Therefore, after replication, DNA is hemimethylated, and DNMT1 adds methyl groups to cytosines in the newly synthesized strand. This process, known as DNA methylation maintenance, also plays a role in restoring spontaneously dissociated methyl groups, with the involvement of histones [6]. In addition to passive demethylation, DNA can undergo active demethylation by the ten-eleven translocation (TET) proteins TET1-3, with the assistance of thymine DNA glycosylase, a base excision DNA repair enzyme [7]. Humans possess another DNMT, DNMT2, which methylates cytosines in both DNA and RNA; however, the properties of this enzyme are not fully understood, making its placement in the epigenetic landscape uncertain [8]. The conversion of C to 5-mC and then to hydroxymethylcytosine, facilitated by the TET proteins, is called DNA hydroxymethylation.

In humans, cytosine is primarily methylated within the 5′-CpG-3′ (CpG) dinucleotides [9]. Some regions of the human genome show an extraordinarily high occurrence of CpG dinucleotides and are known as CpG islands. These CpG islands are frequently associated with gene regulatory regions, including promoters and enhancers, and usually remain unmethylated under normal conditions [10]. Beyond the DNA methylation within CpG islands, DNA can also be methylated through RNA-directed processes, collectively termed RNA-directed DNA methylation (RdDM), where non-coding RNA molecules guide the addition of methyl groups to specific DNA sequences [11]. So far, the RdDM pathway has been most extensively characterized in plants, though other mechanisms of RNA-directed chromatin modification have also been identified in animals. These mechanisms will not be addressed in this review due to insufficient data regarding their potential in PD.

Although sequencing the human genome and mapping many genes were essential steps in the Human Genome Project, the DNA sequence of a gene typically provides little insight into the phenotype that can arise from that sequence, as all nucleated human cells contain nearly identical DNA. A cell’s identity and functions are determined by gene expression, which depends on the epigenetic modifications imposed on these genes. In addition to DNA methylation and demethylation, these modifications include post-translational histone modifications and alterations induced by non-coding RNAs.

The exact mechanism by which DNA methylation regulates gene expression is not fully understood. For a long time, DNA methylation has been primarily regarded as a means to inhibit gene expression. Methyl groups in the major groove of DNA can sterically hinder the binding of transcription factors and the activation of transcription. Methylated DNA can be specifically recognized by a set of proteins known as methyl-CpG-binding proteins (MBPs) that translate DNA methylation signals into appropriate functional states [12]. In particular, MBPs may recruit additional proteins that prevent the binding of transcription factors (Figure 2). However, the sites of DNA methylation located downstream of the transcription start site provide valuable information about gene expression. Moreover, DNA methylation in the first intron of certain specific genes and cell types has been associated with gene expression [13]. DNTMs involved in DNA methylation can also methylate histone tails in the chromatin region where the gene is situated. Above all, any epigenetic modification should be considered within the context of all other modifications that may interact with one another.

In addition to cytosine DNA methylation, the epigenetic regulation of gene expression also involves post-translational histone modification and the action of regulatory ncRNAs. DNA methylation and post-translational histone modifications interact during development [6]. Histone methylation can help direct DNA methylation patterns, and DNA methylation appears to serve as a template for reconstructing histone modification patterns after DNA replication. Also, histone deacetylation can direct and modulate DNA methylation, but in general, the complex interaction of DNA methylation with epigenetic modifications of chromatin is not subjected in this review.

The precise regulation of DNA methylation is essential for the development and normal functioning of many human organs, and its impairment may lead to pathological symptoms [14]. Several studies report changes in the DNA methylation patterns in human gingival fibroblasts (HGFs) and other cells in periodontal disease (PD).

## 3. The Potential of DNA Methylation in Periodontal Disease

Chronic low-grade inflammation remains a hallmark in the DNA methylation pattern of the human genome, as demonstrated in a multi-ethnic epigenome-wide association study [15]. Therefore, inflammatory genes are natural candidates for DNA methylation in PD. A 2014 study focused on immune–inflammatory gene transcription marked the first high-throughput analysis of DNA methylation in PD [16]. This study revealed that variations in DNA methylation between PD patients and controls are more pronounced in the genes associated with immune–inflammatory processes, particularly in the region near the transcription start site. The findings suggested that DNA methylation in PD could influence chromatin status and regulate the activation or silencing of immune–inflammatory genes, which are crucial in PD pathogenesis. However, the evidence regarding chromatin structure involvement was limited. Additionally, this research indicated an upregulation of the enzymes responsible for DNA methylation and demethylation, DNMT1 and TET1, respectively, in the periodontitis tissue samples.

The DNA methylation of inflammatory-associated genes was examined in a subsequent study assessing the DNA methylation patterns in PD patients undergoing therapy [17]. Conventional periodontal therapy included full-mouth scaling and root planing with ultrasonic and manual instruments, along with 0.2% chlorhexidine, recommended twice daily for 20 days. Four genes were studied: tumor necrosis factor (TNF), mitochondrially encoded cytochrome C oxidase II (COX2), long interspersed nuclear element-1 (LINE-1), and interferon-gamma (IFNG), but only COX2 showed a reduction in DNA methylation patterns in gingival biopsy samples in response to periodontal therapy. It was concluded that the therapy reset the DNA methylation status of the COX2 gene to normal levels, as the baseline DNA methylation was nearly twice as high as in healthy subjects. Therefore, the DNA methylation pattern of the COX2 gene may serve as an early marker of the efficacy of periodontal therapy.

A detailed and comprehensive systematic review from 2020 concluded that there is some evidence, although inconsistent, for an association between PD and DNA methylation in certain genes [18]. The candidate genes included interleukin 6 (IL6), IL6 receptor (IL6R), IFNG, prostaglandin–endoperoxide synthase 2 (PTGS2), suppressor of cytokine signaling 1 (SOCS1), and TNF. However, the final conclusion was that the evidence supporting the involvement of these genes in PD pathogenesis and their potential response to periodontal therapy were too weak. The authors supported their thesis by highlighting substantial variations in reporting sample sizes, patient characteristics, statistical analyses, and study design and outcomes. Therefore, it is important to determine whether subsequent studies have produced results that might alter this conclusion.

Changes in the expression of genes involved in immune and methyltransferase pathways were observed in a high-throughput study [19]. A total of 8029 differentially expressed CpG sites were annotated to the promoters of 4940 genes, including 295 immune genes that were enriched. This study identified CpG sites in 23 differentially co-expressed immune gene promoter regions, including 13 hyper-methylated CpG sites in healthy group samples, while some were methylated in most patients. Five CpG sites were identified as strong markers for periodontitis. Although the authors claimed their results provided new diagnostic features for periodontal disease (PD) and might contribute to personalized PD therapy, high-throughput-based routine PD diagnosis and management is not an immediate prospect. Another high-throughput study identified 43,631 differentially methylated positions (DMPs) in PD patients and controls, along with 536 DMPs in gingivitis and controls [20]. One hundred and twenty-six differentially expressed genes (DEGs) were detected between PD patients and controls, with twelve being differentially methylated between PD and gingivitis. This study helped in identifying genes whose expression might be epigenetically regulated in periodontitis.

A 2022 scoped review assessed 30 studies and concluded that DNA methylation and histone modifications may dampen or promote the periodontal inflammatory response to bacterial challenge [21]. However, this review emphasized the perspective of new epidrugs in PD over methodology, research strategies, and the consistency of the analyzed works.

The Gene Expression Omnibus database provided information on gene expression and methylation in 12 periodontitis samples and 12 normal samples [22]. A total of 668 differential methylated genes (DMGs) were detected, comprising 621 hypo-methylated and 47 hyper-methylated genes, primarily involved in intracellular signaling pathways, cellular components, cell–cell interactions, and cellular behaviors. The hypo-methylated genes were mainly enriched in the cyclic guanosine monophosphate-cGMP-dependent protein kinase 1 (cGMP–PKG), RAF proto-oncogene serine/threonine-protein kinase–mitogen-activated protein kinase 1 (RAF-MAP), and phosphatidylinositol 4,5-bisphosphate 3-kinase catalytic subunit delta isoform–RAC-alpha serine/threonine-protein kinase (PI3K–AKT) signaling pathways, while the hyper-methylated genes were predominantly enriched in the pathways of bacterial invasion of epithelial cells, sphingolipid signaling, and DEG-mediated attractive signaling. A bioinformatic analysis identified a protein–protein interaction network containing 630 nodes and 1790 interactions. Additionally, the top 10 hub genes, serving as central nodes and important for the immune–inflammatory response, were identified. This work confirmed the role of DNA methylation and, consequently, epigenetic modifications in the immune and inflammatory responses in periodontal disease (PD) pathogenesis and highlighted some signaling pathways and genes/proteins that could be considered for future therapy in PD.

Periodontal disease is an independent risk factor for cardiovascular disease (CVD) [23]. Importantly, the increased CVD risk persists, even after mitigating periodontal disease (PD) [24]. It was hypothesized that PD might alter the epigenetic pattern of hematopoietic stem cells in the bone marrow (BM) and that these changes would remain following the clinical elimination of PD, continuing to elevate CVD risk [25]. A BM transplant approach was employed to simulate the clinical elimination of periodontitis and the persistence of the hypothesized epigenetic reprogramming. Mice with a low-density lipoprotein receptor knockout were orally inoculated with *P. gingivalis*. Bone marrow was transplanted into recipient mice, which developed greater atherosclerosis associated with increased cytokine/chemokine levels compared to recipients of BM from sham-inoculated donors. This suggested the mobilization of BM progenitor cells. Three hundred seventy-five differentially methylated regions (DMRs) and global hypo-methylation were observed in recipients of BM from *P. gingivalis*-inoculated donors. Some DMRs were linked to genes involved in DNA methylation/demethylation, and validation showed an increased activity of TET-2 along with a decreased activity of DNTMs. Additional associations between PD and CVD mediated by DNA methylation were also noted. This work demonstrates an important mechanism underlying the association of PD with CVD and proposes a potential element in the pathogenesis of PD. Furthermore, it suggests a prospective stem cell-based therapeutic strategy for PD and CVD.

In summary, some studies report associations between the occurrence of periodontal disease (PD) and DNA methylation of various genes. In most cases, these genes are related to the immune system and inflammatory response, although the involvement of other genes is also noted. Increased levels of DNA methylation in the promoters and other regulatory elements of genes that the host uses to combat pathogen invasion contribute to the development of the pathological phenotype. DNA methylation of certain genes has been linked to the severity of periodontal lesions. However, several of these studies have notable limitations that make comparing their results challenging. High-throughput studies identified tens of thousands of differentially methylated genes (DMGs) in PD patients and healthy controls, underscoring DNA methylation’s role in PD pathogenesis and highlighting its importance. Therefore, targeting the DNA methylation pattern in immune–inflammation-related genes to activate these genes may present a therapeutic option for PD. The epigenetic profile plays a crucial role in the differentiation of stem cells, and this significance is reflected in the importance of DNA methylation in mesenchymal stem cells within PD pathogenesis. Stem cell-based regenerative therapy for PD has proven effective in several randomized clinical trials, but targeting the DNA methylation pattern in dental mesenchymal stem cells could be a novel therapeutic strategy for PD [26].

## 4. Folate: An Essential Element for DNA Methylation and Periodontal Disease

DNA methylation depends on the addition of methyl groups to DNA molecules. S-adenosylmethionine (SAM) serves as a universal donor of the methyl group in methylation reactions, including DNA methylation [27].

SAM is produced through the adenosylation of methionine, which is generated by the remethylation of histidine in folate-mediated one-carbon metabolism (Figure 3) [28]. Hence, folate, a B9 vitamin, serves as a key regulator of methyl group donation in DNA methylation. While folate is the natural source of vitamin B9, folic acid (FA) is its synthetic counterpart used in supplements and fortified food products.

The role of folate in PD pathogenesis, prevention, and therapy has been addressed in several studies that did not analyze folate dietary intake. A cross-sectional study based on the National Health and Nutrition Examination Survey (NHANES) from 1999 to 2004 reported that PD patients with moderate to severe disease had lower cis-β-carotene levels in their blood across all racial and ethnic groups, and these decreased levels worsened with age. Folate differences were significant among various age groups, with reliably lower levels in PD patients older than 30 years, and were most pronounced in females. Lower levels of vitamin D were consistently observed across the entire age range of patients, with a greater difference seen in females with periodontitis [29]. Additionally, an interaction between folate and race was noted among PD patients. Thus, this study, the first large population study on folate in PD, revealed not only a negative correlation between folate and the occurrence and severity of PD but also indicated that blood levels of folate might depend on age, ethnicity, and sex, all of which should be considered in studies on the role of folate in PD pathogenesis.

As folate metabolism is influenced by aging, a population-based cross-sectional study on the serum levels of folic acid in older individuals (aged over 60) based on NHANES was warranted [30]. A negative correlation between periodontal disease and serum folic acid levels was observed in more than 800 subjects, even after controlling for demographics, educational level, body mass index, bleeding on probing (BOP), probing sites, levels of vitamin B12 and homocysteine, chronic diseases, smoking, and alcohol use. This correlation was not affected by sex. Therefore, the clear conclusion from this study was that low serum folic acid levels were independently associated with periodontal disease in older adults. These results suggest that serum folate levels are important indicators of periodontal disease in the elderly, and, therefore, folic acid may be a target for promoting oral health.

A recent cross-sectional study based on NHANES 2009–2014 found negative associations between serum folate and periodontitis status, periodontal probing depth, and clinical attachment loss [31]. A negative association was also identified between red blood cell folate and periodontitis status. These correlations were noted in adults aged 30 years and older. The negative correlation between folate levels in red blood cells and periodontal status was confirmed in another recent NHANES-based study [32].

The intracellular action of folate is mediated by its alpha (FOLR1) and beta (FOLR2) receptors. Therefore, the biological effects of folate depend not only on its concentration but also on the availability of its receptors. It was noted that FOLR1 levels in the gingival crevicular fluid (GCF) were higher in patients with gingivitis and periodontitis compared to controls [33]. A significant correlation was found between GCF FOLR1 levels and POB. However, no differences were observed between PD patients and controls regarding serum FOLR1 levels. Thus, that study indicated that the expression of folate receptors may be stimulated by processes involved in PD pathogenesis in the target tissue. Such localized overexpression of the FOLR1 gene has been observed in many solid cancers, making it a therapeutic target in oncology [34].

These studies have significant limitations because the dietary intake of folate was not controlled or included in the analyses. Furthermore, folate levels vary with age and may depend on ethnicity, which should be considered as a confounding factor.

## 5. The Potential of a Folate-Rich Diet for Targeting DNA Methylation in Periodontal Disease

Nutrition and oral hygiene are closely linked to oral health. Reducing sugar in the diet and maintaining appropriate vitamin levels have positive effects on the prevention and treatment of PD [35]. Oxidative stress is associated with PD; therefore, a diet rich in antioxidants may be helpful in managing it. However, several studies regarding the impact of antioxidant-rich diets on PD show no significant effects or results that can be replicated. This may be because most antioxidant-enriched diets include vitamins A, C, and E. The cellular antioxidant system consists of three components: antioxidant enzymes, DNA repair proteins, and low molecular weight antioxidants, but the importance of the latter is much less than that of the former. Therefore, any diet thought to have antioxidant properties should be assessed for its potential to influence antioxidant enzymes and DNA repair mechanisms. The role of folate in oxidative stress will be discussed in the next section.

In addition to its critical role in one-carbon metabolism, folate is an essential micronutrient that humans must obtain from their diet. Folate supplements are available in the forms of folic acid, folinic acid, or 5-methyltetrahydrofolate (5-MTHF). A folate deficiency is linked to several diseases, including cardiovascular disease (CVD) and cancer, although the mechanism underlying this association is not fully understood [36,37]. However, excessive folate intake may also be harmful to individuals with cancer or CVD [38]. Thus, these two groups of diseases may be particularly sensitive to any deviation from the required level of folic acid. As mentioned in the previous section, folic acid levels in serum, red blood cells, and the gingival crevicular fluid (GCF) of patients with periodontal disease (PD) may differ from those in control subjects. Consequently, it is reasonable to investigate the relationship between folic acid intake and the occurrence and progression of PD. Dietary folic acid intake is associated with the distribution of differentially methylated positions and regions in large-scale, epigenome-wide association studies [39]. Therefore, alterations in dietary folic acid content may change the DNA methylation patterns.

Several studies indicate a negative correlation between dietary fatty acid intake and the occurrence and severity of periodontal disease. A study from 1976 found that dietary supplementation with fatty acids improved the resistance of the gingiva to local irritants and reduced inflammation [40]. A negative correlation was also observed between dietary fatty acid levels and bleeding on probing in a Japanese population of non-smokers [41]. However, that study did not find any significant association between community periodontal index scores and fatty acid intake levels.

A study using data from NHANES between 2011 and 2014 found that insufficient intake of folate was positively correlated with the severity of PD [42].

Systemic folate intake was evaluated for its clinical and biochemical effects in scaling and root planing (SRP) for the treatment of 60 PD patients [43]. Half of the participants supplemented their diet with FA, while the other half received a placebo. Both groups exhibited a time-dependent reduction in plaque index (PI), gingival index (GI), probing pocket depth (PPD), clinical attachment level (CAL), and gingival recession (GR). The study indicated that dietary supplementation with folate/FA might enhance the outcomes of PD treatment.

Several other studies suggest a link or lack thereof between folate intake and PD. Woelber et al. examined randomized clinical trials regarding dietary interventions, including folate intake, and their effects on PD clinical outcomes [44]. They highlighted several limitations of these studies, such as inadequate reporting of within-group differences, the inability to blind participants to dietary changes or the consumption of specific foods, and challenges in determining the actual impact of dietary interventions alongside non-surgical PD therapy.

The exact mechanism(s) by which folate may exert a beneficial effect in PD is not known, although it is suggested that this is related to the antioxidant properties of folate. This will be discussed in the subsequent section, also in the context of DNA methylation. However, we now want to consider the suitability of dietary intervention for specifically modifying DNA methylation or, more generally, the epigenetic pattern on a global scale or in specific cells.

Similar to the genome, the epigenome can be influenced by dietary compounds; however, diet-induced changes in the genome are significantly less likely than those in the epigenome [45]. Genetic alterations that do not stem from DNA metabolism are addressed by evolutionarily developed cellular DNA damage response (DDR). Currently, there is no known system to repair or reverse epigenetic changes. Therefore, it may be simpler to alter the epigenome through diet rather than the genome; nonetheless, controlling the changes in the epigenome resulting from external actions is challenging for several reasons. Firstly, the epigenetic profile is shaped by the activity of numerous enzymes, proteins, and RNAs involved in DNA methylation/demethylation, post-translational modifications of histones, and the action of non-coding RNAs. These three components of the epigenetic profile interact, meaning that altering one component may lead to changes in the others that can be difficult to detect [46]. Additionally, unlike genetic modifications that are confined to specific sites, the possible combinations of chemical modifications to all histone tails are extraordinarily vast. However, many drugs targeting proteins involved in establishing and maintaining the cellular epigenetic pattern have either been approved for therapy or are under investigation in clinical trials, particularly in cancer treatment [47]. Epigenetic therapies primarily focus on DNA methylation and the chemical modifications of histones [48].

Chemicals that can alter the epigenetic profile may be components of a normal or epigenetically focused diet, but any diet can introduce changes to the epigenetic profile. Moreover, epigenetic drugs are administered alone or in combination with a limited number of other chemicals; however, administering a chemical as a food component may affect its bioavailability and action. Therefore, the consequences of an epigenetic diet may be harder to predict than those of epigenetic drugs. Accordingly, the term “epigenetic diet” is not entirely justified [49,50,51].

In summary, while evidence regarding the role of folic acid in preventing PD and improving its outcomes is emerging, as is the evidence concerning the role of epigenetic modifications of genes, it is not rational to project dietary interventions with folate targeting DNA methylation since it may be unrealistic. However, it cannot be ruled out that the mechanisms underlying the effects of folate in PD involve changes in DNA methylation.

## 6. Antioxidant Effects of Folate on Periodontal Disease

As shown in Figure 3, folate is indirectly linked to homocysteine production in the methionine cycle. Homocysteine is an independent risk factor for neurodegenerative diseases and CVD, but the protective role of folate extends beyond merely reducing homocysteine; it also encompasses antioxidant properties [52]. The antioxidant potential of folate may underlie its beneficial effects in PD, as antioxidants are known to positively influence adjuvant treatments for PD [53]. While it is challenging to identify a disease without oxidative stress in its etiology, consistent reports highlight the role of oxidative stress in PD pathogenesis [54]. Oxidative stress is connected to the production of reactive oxygen and nitrogen species (RONS), which are not effectively neutralized by antioxidant defense systems. RONS can damage biological macromolecules, including proteins, lipids, and DNA. However, DNA occupies a unique position, as cells possess a rich array of mechanisms for addressing DNA damage, known as the DNA damage response (DDR), which includes several methods for repairing DNA. Therefore, if DNA damage occurs, an immediate question arises regarding its repair. Although DNA damage in PD has been discussed in numerous papers, information specific to DNA repair in this disease remains limited. A 2021 systematic review and meta-analysis indicated that folate supplementation might enhance total antioxidant capacity and levels of reduced glutathione (GSH), while also reducing concentrations of malondialdehyde (MDA) [55].

The total antioxidant capacity in the GCF of PD patients was lower than that in controls [56]. Similar differences were noted in plasma and serum levels. Therefore, PD may be linked to local and systemic impairment of antioxidant defense due to predisposition to PD or systemic low-grade inflammation induced by invading bacteria, or it may be an inherent characteristic of PD patients. Similar studies have shown a positive association between decreased TOC in serum and the severity of PD [57]. In general, numerous animal or in vitro experiments and interventional studies in patients suggest that antioxidant strategies could effectively prevent and treat PD [58]. However, the exact source of stress-related RONS is not clearly identified; the main culprits are likely the neutrophils and macrophages of the host’s innate immune system, as well as invading bacteria that produce RONS [54,59,60].

As mentioned, there are three main elements of antioxidant defense: antioxidant enzymes and peptides, DNA repair proteins, and low molecular weight antioxidants. DNA methylation may directly regulate the expression of genes encoding antioxidant enzymes. The stimulation of genes coding for DNA repair proteins may result in a higher efficacy in removing oxidative DNA damage. It may also enhance repair, thus increasing the functionality of genes encoding antioxidant enzymes.

Chronic inflammation typical of PD may also be a source of DNA damage. However, the interplay between oxidative stress and inflammation is complex since oxidative stress may cause inflammation and vice versa [61]. Similar to oxidative stress, inflammation-related DNA damage can be caused by RONS produced during inflammatory processes. There are many sources of RONS in inflammation, with polymorphonuclear neutrophils being a leading producer at the site of inflammation [62]. The highly cited article, “Periodontitis is an inflammatory disease of oxidative stress: We should treat it that way,” adequately reflects the significance of the complex relationship between oxidative stress and inflammation in PD pathogenesis [63]. Therefore, oxidative stress as a factor in PD pathogenesis should be considered alongside inflammation.

PD is frequently associated as a comorbidity with diseases where oxidative stress plays an important role in their pathogenesis, including CVD, type 2 diabetes mellitus (T2DM), certain neurodegenerative diseases, and some cancers [64]. These comorbidities are characterized by DNA damage either as an associated factor (e.g., T2DM) or as an important element of pathogenesis (e.g., cancer) [65,66]. It was observed that changes in the activity of superoxide dismutase 2 (SOD2) in PD resulted in an elevation of the ACHT, LRR, and PYD domain-containing protein 3 (NLRP3) inflammasome–caspase-1–interleukin 1β axis in human NIH3T3 fibroblasts [67]. This work illustrates the interplay between oxidative stress and inflammation in PD pathogenesis, and the authors interpreted the results as a limitation of the defense inflammatory reaction to prevent too much harm to the host. Epidemiological studies indicate that the risk of periodontitis in diabetes is about three times higher than in the general population (diabetes mellitus-associated periodontitis) [68]. Moreover, a positive correlation between hyperglycemia and the intensity of periodontic symptoms has been observed [69]. Hyperglycemia can stimulate as many as eight types of oxidases that may generate oxidative stress [70].

Can the antioxidant mechanism of folate be related to its involvement in DNA methylation? The immediate answer is yes. If folate is involved in the epigenetic regulation of gene expression, it may alter the expression of genes involved in redox regulation. Oxidative stress is a complex state with many factors necessary for its occurrence. The overproduction of RONS associated with oxidative stress may be caused by various factors, including the deregulation of genes involved in antioxidant defense. RONS overproduction may result from the overactivity of some proteins, such as proteins in the mitochondrial electron chain (ETC) that produce RONS, even during normal functioning. The downregulation of genes encoding such overactive proteins would lead to reduced RONS levels. Therefore, to be considered an antioxidant, a compound does not necessarily need to be a direct neutralizer of RONS but may contribute to reducing RONS production. Whether folate exerts antioxidant effects through the DNA methylation-mediated downregulation of genes whose products induce RONS remains an open question. However, it is worthwhile to examine the role of mitochondria in PD pathogenesis, as these organelles are the main RONS producers in humans.

## 7. Mitochondria in the Pathogenesis of Periodontal Disease

Mitochondria are the primary energy producers in humans, converting substrates from food into ATP through oxidative phosphorylation (OXPXOS). ATP serves as a substrate for chemical reactions that supply the energy necessary for cellular processes required for maintaining organismal homeostasis.

OXPHOS refers to a series of reactions that take place in the mitochondrial electron transport chain (ETC), comprising five protein complexes. Even the normal functioning of the ETS leads to the production of reactive oxygen and nitrogen species (RONS), and mitochondrial dysfunction can exacerbate this effect. Thus, it is not surprising that mitochondria evolved separately from the nucleus, which serves as the primary reservoir of genetic information. Mitochondria possess their own mitochondrial DNA (mtDNA, the mitochondrial genome), encoding two rRNAs, twenty-two tRNAs, and thirteen polypeptides, all of which are subunits of the ETS: seven subunits of complex I (NADH: ubiquinone oxidoreductase), one subunit of complex III (ubiquinol–cytochrome c oxidoreductase), three subunits of complex IV (cytochrome c oxidase), and two subunits of complex V (ATP synthase) (Figure 4). All other subunits of the ETC complexes are encoded by the nuclear genome. Due to RONS production and their involvement in inflammatory reactions, mitochondria may play a role in the pathogenesis of PD. This potential is further supported by the significant role of mitochondria in apoptosis, which is also implicated in PD pathogenesis [71].

Mitochondrial homeostasis is upheld by mitochondrial quality control (mtQC), which coordinates essential mitochondrial processes such as proteostasis, biogenesis, dynamics, and mitophagy to protect mitochondria from damage and prevent the accumulation of harmful lesions that could perpetuate the mitochondrial vicious cycle [72]. Although RONS production is regarded as the primary damaging factor within mitochondria, RONS are not harmful as long as they remain under the control of mtQC, whose impairment is linked to numerous human diseases [73].

Multiple studies indicate a connection between PD and mitochondrial dysfunction. Various research efforts have been made to investigate the correlation between mtDNA variability and the incidence and severity of PD (reviewed in [74]). In general, numerous mtDNA mutations have been linked to PD across several studies; however, some of these studies exhibit significant limitations, including a small sample size and a lack of analysis of confounding factors related to both dependent and independent variables. It is important to emphasize that some studies demonstrated the specificity of mtDNA mutations in PD for targeted tissues rather than for peripheral blood [75]. Additionally, certain studies identifying mtDNA mutations as exclusive to PD are less convincing, as these mutations have also been linked to a range of other conditions, such as mitochondrial diseases, stroke, type 2 diabetes mellitus (T2DM), leukemia, and multiple sclerosis [76]. Many mtDNA mutations and single-nucleotide polymorphisms (SNPs) have been associated with the more aggressive forms of PD [77,78]. Phenotypic manifestations of mitochondrial dysfunction in PD include impaired oxidative phosphorylation (OXPHOS), attributed to decreased expression of OXPHOS proteins, reduced mitochondrial membrane potential (MMP), increased reactive oxygen and nitrogen species (RONS) production, a lower cell proliferation rate, a diminished level of coenzyme Q10 (CoQ10), and decreased citrate synthase activity [79,80,81]. These findings have been corroborated by animal studies and in vitro research on oral mucosal fibroblasts from PD patients [82,83].

Several works examine the role of invading bacteria in disrupting mitochondrial homeostasis (reviewed in 85,86). Lipopolysaccharide (LPS), released by various bacterial species involved in PD pathogenesis, is a natural candidate for causing damage to periodontal tissue. LPS from *P. gingivalis* induced mitochondrial abnormalities in normal human gingival fibroblasts (HGFs) similar to those observed in HGFs from PD patients [83]. Additionally, LPS is not the only virulence factor of *P. gingivalis* that may impact mitochondrial homeostasis. Studies indicated that *P. gingivalis* prompted a shift in macrophage metabolism from oxidative phosphorylation (OXPHOS) to glycolysis, while macrophages and T helper 17 (Th17) cells primarily using OXPHOS exhibited anti-periodontitis properties [84,85]. Furthermore, these cells resisted apoptosis, unlike their counterparts under glycolytic conditions [86]. The transition from pro-inflammatory M1 macrophages to their anti-inflammatory M2 counterparts involved a shift from glycolysis to OXPHOS and facilitated bone formation in damaged canine periodontal tissue defects [87]. The change from OXPHOS to glycolysis is linked to an increase in mitochondrial membrane potential (MMP). Therefore, the conversion between glycolysis and OXPHOS may play a crucial role in PD pathogenesis.

The exact mechanism by which changes in mitochondrial homeostasis contribute to inflammation induced by *P. gingivalis* remains unclear [42]. Many mitochondrial components and metabolites can act as damage-associated molecular patterns (DAMPs) that promote inflammation when released into the cytosol or extracellular medium [88]. mtDNA is one of the most well-recognized DAMPs, and it can be released from mitochondria into the cytosol through pathways that regulate the permeability of the mitochondrial membrane, including the mitochondrial permeability transition pore (mPTP) in response to mitochondrial insult, leading to the activation of immune-related pathways [89]. These pathways include aberrant pro-inflammatory responses and type I interferon (IFN) responses triggered by the recognition of released mtDNA by various pattern recognition receptors such as NLRPs, toll-like receptors (TLRs), and cyclic GMP/AMP synthase, and the stimulator of interferon gene (STING) systems [42]. This may stimulate mitochondria-mediated apoptosis in mouse osteoblasts and human gingival epithelial cells [90,91]. However, some findings suggest that apoptosis may not be an immediate response in periodontal disease (PD), as an upregulation of anti-apoptotic proteins, including the apoptosis regulator Bcl-2, has been observed in the gingival tissue and oral neutrophils of PD patients [92,93]. It is concluded that delayed apoptosis may lead to the progressive destruction of periodontal tissue.

Bacteria invading periodontal tissue may influence several aspects of mtQC. The LPS from *P. gingivalis* has been shown to disrupt two key mtQC regulators: peroxisome proliferator-activated receptor gamma coactivator alpha (PGC-1α) and mitochondrial transcription factor A (TFAM) in HGFs [94]. HGFs exposed to LPS exhibited fewer mtDNA copies; however, mitochondrial biogenesis improved with antioxidant treatment. The reports on the involvement of invading bacteria in mitochondria-mediated apoptosis in the periodontal cells are inconsistent. Both direct induction of apoptosis and its suppression, likely to delay it, are reported, and details can be found elsewhere [42].

Thus, PD may be linked to various pathways of mitochondrial dysfunction. Some of these are illustrated in Figure 5. All these reactions might be triggered by pathogens invading periodontal tissue, though some may indeed occur as a secondary effect of the invasion. The issue of apoptosis caused by mitochondrial dysfunction in PD necessitates further investigation, as it can represent a double-edged sword in the relationship between the pathogen and host.

Although the epigenetic modification of mtDNA is an emerging area of study, many aspects of this issue still await explanation and remain subjects of debate, with some controversial findings [95]. Due to the absence of nucleosomal organization, mtDNA is susceptible to only two main classes of epigenetic modifications: DNA methylation and modifications caused by ncRNAs. However, similar to genetic variability, the high number of mtDNA copies allows for an enormous array of possible epigenetic modifications that can be imposed on the mtDNA of a single cell. While there are no CpG islands in mtDNA like those found in the nuclear genome, dispersed CpG dinucleotides do occur in mtDNA. However, the biological significance of mitochondrial CpG dinucleotides is lower than that of their nuclear counterparts, and mtDNA is extensively methylated in non-CpG sequences [96,97]. Methyl groups attached to mtDNA are primarily provided by mitochondrial DNA methyltransferase 1 (mtDNMT1), an isoform of nuclear DNMT1. Several proteins that are crucial for maintaining the epigenetic profile in nuclear DNA have been identified in mitochondria, including DNMT1, DNMT3A, DNMT3B, and TET1/2. However, the existence and mechanisms of epigenetic modifications to mtDNA remain unresolved issues [98,99,100,101].

Folate-mediated one-carbon metabolism in human cells is compartmentalized into the cytosol, nucleus, and mitochondria [102]. Folate metabolism in mitochondria parallels its cytosolic pathway, as these two pathways are interconnected by the transport of one-carbon sources. One of the most significant discoveries in understanding folate metabolism was the identification of mitochondria-specific, folate-dependent thymidylate (deoxythymidine monophosphate, dTMP) synthesis. Any imbalance in the dTMP pool compromises DNA integrity, both nuclear and mitochondrial, and mice deficient in thymidine kinase 2 exhibited a progressive loss of mtDNA. Impaired mtDNA integrity is manifested through deletions that may lead to the silencing of genes essential for mitochondrial homeostasis and may be related to several human mitochondrial diseases. Many mitochondrial functions are compromised due to impairment in folate metabolism within that organelle.

In summary, there is an interplay between folate metabolism, PD pathogenesis, and mitochondrial functions (Figure 6). Folate may benefit PD by directly scavenging RONS, enhancing mitochondrial functions, and modulating the expression of genes involved in oxidative stress. The question of whether these genes also include mitochondrial genes cannot be answered at this time. While the genes encoding ETC subunits can be considered, other genes that encode non-coding RNAs cannot be directly linked to oxidative stress or RONS scavenging. The role of folate as a coenzyme in mtDNA methylation should be further explored, along with comprehensive studies on the mechanisms and significance of epigenetic modifications to mtDNA.

## 8. Conclusions, Key Questions, and Perspectives

Folate may benefit periodontal disease (PD) by directly scavenging reactive oxygen and nitrogen species (RONS) produced during the immune–inflammatory response to the invasion of periodontal pathogens. Additionally, folate may enhance mitochondrial functions that become compromised during the progression of PD. A third mechanism contributing to the beneficial effects of folate in PD pathogenesis involves modulating the expression of genes that may lead to increased RONS production through the methylation of their regulatory sequences. These conclusions are based on experimental evidence and suppositions. The question of whether DNA methylation is restricted to nuclear DNA or can also include its mitochondrial counterpart remains unresolved, as our understanding of DNA methylation in mitochondrial genes and the roles of tRNAs and rRNAs encoded by the mitochondrial genome in oxidative stress and RONS production is still insufficient.

Epigenetic modifications of nuclear DNA, including DNA methylation, are reported to be associated with the occurrence and severity of periodontal disease (PD). These studies primarily describe changes in the target, periodontal tissue, but some also indicate systemic alterations. The concept of end-organ deficiency posits that different concentrations of a substance can exist in various tissues, despite a normal plasma concentration. This concept has been applied to the inflammatory changes in the gingivae induced by the topical or systemic administration of folate/FA. Therefore, one might conclude that the topical administration of FA could be more effective in preventing PD than corresponding dietary interventions. However, the preventive topical administration of PA may be burdensome for patients and ineffective. Dietary intervention by adjusting the diet with folate-rich foods or supplementing with FA-fortified nutrients, along with antioxidants, makes the results difficult to predict due to the interaction between folate/FA and other dietary components. Conversely, while the topical administration of FA in PD is easier to manage, it involves a frequent and burdensome repetitive procedure. Therefore, additional studies on the folate/FA delivery route in PD are necessary.

Aging and smoking are the primary risk factors for PD, and as such, they should be addressed in all human studies related to PD. Therefore, research on the role of folate in PD pathogenesis must consider that smoking increases the risk of PD and reduces folate concentration. Consequently, smoking may serve as a link between PD and folate deficiency. However, this is not the only connection, as PD can also occur in non-smokers. Nevertheless, the relationship between smoking, PD, and folate warrants further investigation, particularly regarding folate supplementation for active smokers affected by PD. Folate has been reported to have certain anti-aging effects, so it is worth exploring whether the beneficial impact of folate on PD intersects with its anti-aging potential.

In this review, we limited our considerations to oxidative stress as the primary mechanism underlying PD pathogenesis. However, oxidative stress has at least two aspects in PD. Invading bacteria may trigger oxidative stress that leads to periodontal damage, but the host may also produce oxidative stress to fight invading pathogens, as reactive oxygen and nitrogen species (RONS) possess antimicrobial properties. Reports indicate that macrophages generate mitochondrial reactive oxygen species (mtROS) to resist bacterial invasion via the direct interaction of activated toll-like receptors (TLRs) with mitochondrial complex I. Therefore, the role of oxidative stress in PD pathogenesis requires further studies to address unresolved questions regarding the complex relationships between the host and invaders.

Another limitation of our study was that it focused exclusively on DNA methylation as a representation of the epigenetic modifications that can affect cells. In fact, DNA methylation is a relatively simple modification compared to histone modifications and the changes brought about by non-coding RNAs. Therefore, alterations in DNA methylation may impact the epigenome, but they may not necessarily change the pattern of gene expression or alter it in a way that deviates from the expected effects stemming solely from DNA methylation.

Consequently, folate-based dietary intervention can be recommended for PD prevention and as adjuvant therapy, especially for individuals with folate deficiency, including smokers and the elderly. Whether DNA, including mtDNA, methylation is part of the mechanism behind the beneficial effects of folate in PD remains to be determined in future research.

## Figures and Tables

**Figure 1 ijms-26-03225-f001:**
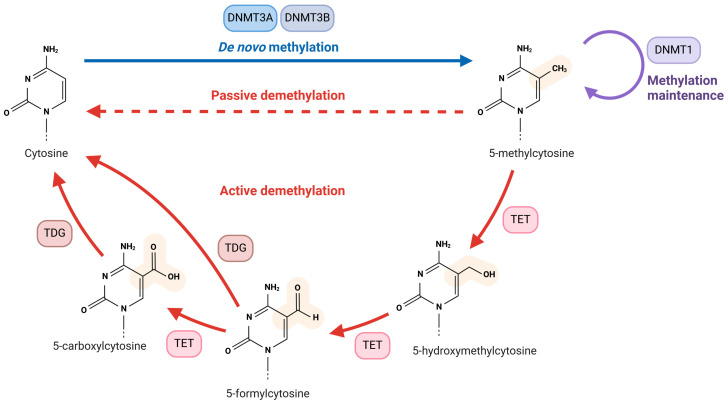
Main stages of DNA methylation and demethylation in humans. Cytosine is methylated by the DNA methyltransferases DNMT3A and DNMT3, which add methyl groups to both DNA strands, producing 5-methylcytosine (5-mC) through de novo methylation. After replication, the newly synthesized strand is non-methylated, resulting in hemimethylated DNA. This is followed by DNMT1 adding the missing methyl group, leading to fully methylated DNA, which is known as DNA methylation maintenance. In addition to passive demethylation, 5-mC can be actively demethylated by ten-eleven translocation (TET) proteins, which convert it into 5-hydroxymethylcytosine. This molecule can be further transformed into 5-formylcytosine and then into 5-carboxylcytosine, both of which can ultimately be converted back to cytosine by thymine DNA glycosylase (TDG), an enzyme involved in DNA repair. This scheme presents the methylation of “naked” DNA and does not include effects that may result from chromatin, including those stemming from the interaction between DNA methylation and histone modification machinery. Created in https://BioRender.com.

**Figure 2 ijms-26-03225-f002:**
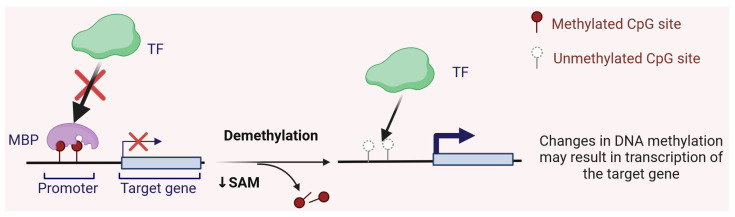
The role of DNA methylation in gene expression regulation. Methylated DNA at the CpG dinucleotides within the gene promoter can be specifically recognized by a group of proteins known as methyl-binding proteins (MBPs), which recruit other proteins not depicted in this scheme. This action blocks transcription factors (TFs) from accessing the promoter, leading to the silencing of the target gene. Conversely, demethylation of the promoter, which can occur through active or passive DNA demethylation or reduced availability of S-adenosylmethionine (SAM)—a universal donor of methyl groups—may activate transcription of the target gene. This scheme does not account for the impact of other epigenetic modifications on the target gene. Created in https://BioRender.com.

**Figure 3 ijms-26-03225-f003:**
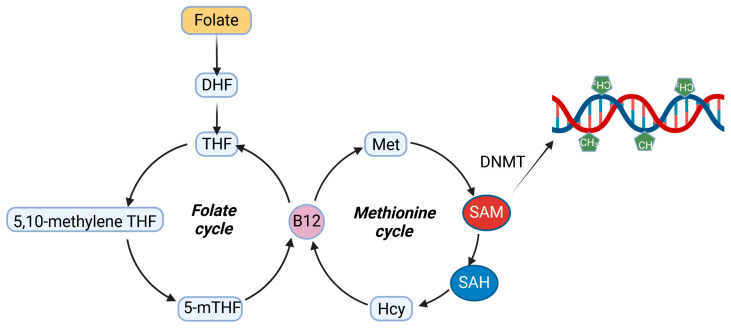
Folate is an essential component in one-carbon metabolism, encompassing the series of reactions in the folate and methionine cycles that produce S-adenosylmethionine (SAM), a universal methyl group donor vital for various molecular processes, including DNA methylation. Vitamin B12 acts as a cofactor in both cycles. DHF—dihydrofolate, THF—tetrahydrofolate, 5-mTHF—5-methyl THF, Met—methionine, SAH—S-adenosylhomocysteine, Hcy—homocysteine, DNMT—DNA methyltransferase. Created in https://BioRender.com.

**Figure 4 ijms-26-03225-f004:**
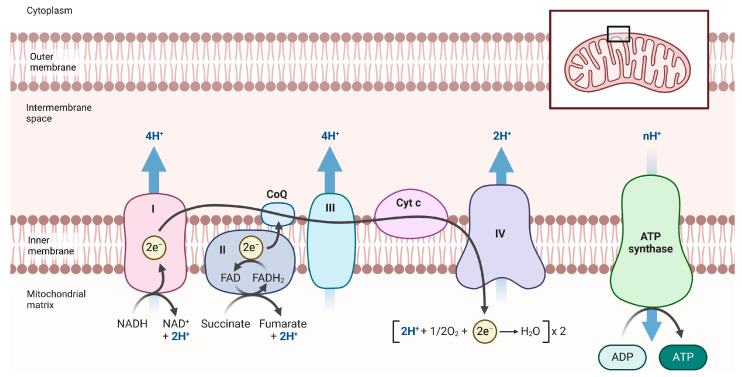
The mitochondrial electron transport chain (ETC) comprises protein complexes anchored to the inner mitochondrial membrane. The tricarboxylic acid cycle in the mitochondrial matrix supplies reduced nicotinamide adenine dinucleotide (NADH) and reduced flavin adenine dinucleotide (FADH2) to the ETC, with each donating a pair of electrons via complexes I and II, respectively. The transfer of electrons from complex I results in the pumping of 4 protons across the inner membrane into the intermembrane space. The electrons from complexes I and II are passed to ubiquinone (CoQ), which is reduced to ubiquinol (QH2) and oxidized by complex III, enabling one electron to travel through cytochrome c (Cyt c). Cytochrome c carries electrons to complex IV, where molecular oxygen is reduced to water. The movement of protons from the mitochondrial matrix into the intermembrane space, driven by electron transfer, establishes the mitochondrial membrane potential, which is dissipated when H+ re-enters the matrix through complex V, coupled with the production of ATP from ADP. Created in https://BioRender.com.

**Figure 5 ijms-26-03225-f005:**
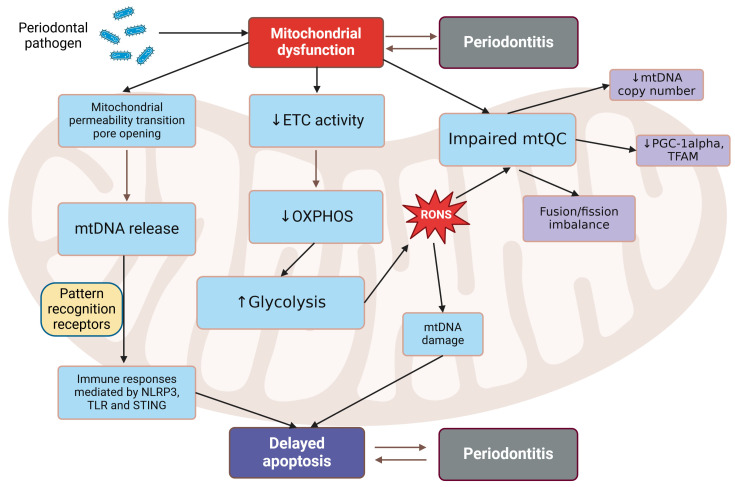
Periodontal pathogens may induce mitochondrial dysfunctions associated with periodontal disease. These dysfunctions can manifest as the opening of mitochondrial pores, which facilitate the release of mtDNA recognized by pattern recognition receptors. This induces inflammatory responses mediated by ACHT, LRR, and PYD domain-containing protein 3 (NLRP3), toll-like receptors (TLR), cyclic GMP-AMP synthase (cGAS), and the stimulator of interferon gene (STING) pathways, potentially contributing to the activation of the mitochondrial pathway of apoptosis. Mitochondrial dysfunctions may involve impairment of the electron transport chain (ETC), leading to depression in oxidative phosphorylation (OXPHOS) and an increase in glycolysis. This results in the overproduction of reactive oxygen and nitrogen species (RONS), which can damage mtDNA and stimulate apoptosis. Impairments in mitochondrial quality control may include the downregulation of key proteins involved in mitochondrial biogenesis, such as peroxisome proliferator-activated receptor gamma coactivator alpha (PGC-1α) and mitochondrial transcription factor A (TFAM), as well as the disruption of fusion/fission balance and changes in the number of mtDNA copies. Created in https://BioRender.com.

**Figure 6 ijms-26-03225-f006:**
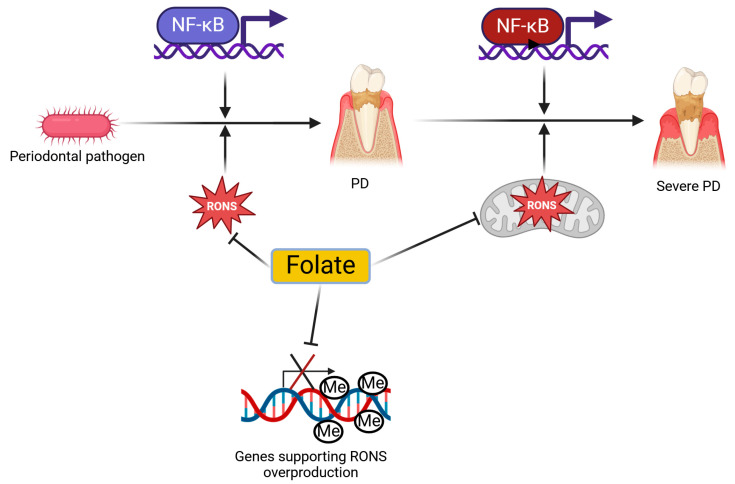
Folate may benefit periodontal disease (PD). The invasion of periodontal pathogens can trigger the immune–inflammatory response, leading to the activation of various proteins and their complexes, including the nuclear factor NF-kappa-B (NF-κB) complex, which modulates the expression of pro- and anti-inflammatory genes. This invasion may be linked to oxidative stress and increased production of reactive oxygen and nitrogen species (RONS), which are directly induced by the pathogens and produced in the host as a result of inflammation. Mitochondrial dysfunctions, often associated with RONS overproduction, contribute to the pathogenesis of PD, and it is hypothesized that this may be particularly significant in the severe forms of the disease. Folate may benefit PD by inhibiting RONS, improving mitochondrial dysfunctions, or downregulating the expression of genes involved in RONS overproduction through the methylation of their regulatory sequences. The different colors of NF-κB represent various genes that may be activated by NF-κB at different stages of PD progression. Created in https://BioRender.com.

## Data Availability

Not applicable.

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
