# Peer review of "DNA Methylation in Periodontal Disease: A Focus on Folate, Folic Acid, Mitochondria, and Dietary Intervention"

_ijms, 2025, doi:10.3390/ijms26073225_

Round 1

Reviewer 1 Report

Comments and Suggestions for Authors

This is a well written narrative review. However it needs some minor changes in order to clarify theoretical concepts.

1) Lines 159-167 and Figure 1 do not consider the relation between DNA methylation and histone acetylation as a mechanism to modulate chromatin remodeling.

2) Line 295: FA acronym must be defined previously as folic acid. In addition, the differences between FA y folates must be mentioned in the text. 

3) Lines 308 and 312: the authors mentioned the concept "circulating leves of FA". Actually, there is no circulating levels of FA but there is circulating levels of folates. When FA is absorbed in small intestine and metabolized by enterocytes, is converted in folates which are detected in blood. The same concept must be applied to line 335 mentioned "intracellular FA). Thus, in the text the concept "FA metabolism" is detected in several lines which is also incorrect. 

Author Response

This is a well written narrative review. However it needs some minor changes in order to clarify theoretical concepts.

Comment: 1) Lines 159-167 and Figure 1 do not consider the relation between DNA methylation and histone acetylation as a mechanism to modulate chromatin remodeling.

Answer: Our review focuses on DNA methylation and the interaction between DNA methylation and chromatin structure is a subject deserving another review. Figure 1 presents mechanisms of DNA methylation, demethylation, and hydroxylation regardless of chromatin. We have added the following sentence in the Figure 1 legend:

“This scheme presents the methylation of “naked” DNA and does not include effects that may result from chromatin, including those stemming from the interaction between DNA methylation and histone modification machinery.”

Also, we have added the following sentence to that section (from line 173 in the original submission):

“Also, histone deacetylation can direct and modulate DNA methylation, but in general the complex interaction of DNA methylation with epigenetic modifications of chromatin is not subjected in this review.”

Comment: 2) Line 295: FA acronym must be defined previously as folic acid. In addition, the differences between FA y folates must be mentioned in the text.

Answer: Indeed! We have added the following fragment to section 4, starting at line 296 in the original submission:

“While folate is the natural source of vitamin B9, folic acid (FA) is its synthetic counterpart used in supplements and fortified food products.”

and we have reserved the acronym FA for folic acid only from that point onward in the text.

Furthermore, we have changed the title of our manuscript from: “DNA Methylation in Periodontal Disease: Focus on Folate, Mitochondria, and Dietary Intervention” to: “DNA Methylation in Periodontal Disease: Focus on Folate, Folic Acid, Mitochondria, and Dietary Intervention.”

Comment: 3) Lines 308 and 312: the authors mentioned the concept "circulating leves of FA". Actually, there is no circulating levels of FA but there is circulating levels of folates. When FA is absorbed in small intestine and metabolized by enterocytes, is converted in folates which are detected in blood. The same concept must be applied to line 335 mentioned "intracellular FA). Thus, in the text the concept "FA metabolism" is detected in several lines which is also incorrect.

Answer: As we mentioned in response to the previous comment, we have corrected the use of "folate," “folic acid," and “FA” throughout the entire manuscript.

Reviewer 2 Report

Comments and Suggestions for Authors

Dear Authors, 

congratulations for the study. 

I would suggest to speculate on how the folate and antioxidants can be delivered to be more effective.

Beyond the nutrition and diet...maybe a local administration would be more effective? 

Author Response

Comment: Dear Authors, congratulations for the study.

Answer: Thank you.

Comment: I would suggest to speculate on how the folate and antioxidants can be delivered to be more effective.

Answer: Please see our answer to your subsequent comment.

Comment: Beyond the nutrition and diet...maybe a local administration would be more effective?

Answer: We have addressed that problem in lines 693-702 in the original submission. Additionally, we have included the following sentence in that paragraph:

“Dietary intervention by adjusting the diet with folate-rich foods or supplementing with FA-fortified nutrients, along with antioxidants, makes the results difficult to predict due to the interaction between folate/FA and other dietary components. Conversely, while topical administration of FA in PD is easier to manage, it involves a frequent and burdensome repetitive procedure. Therefore, additional studies on the folate/FA delivery route in PD are necessary.”